# GENERALIZATION BOUNDS
# WITH ARBITRARY COMPLEXITY MEASURES

## ABSTRACT

In statistical learning theory, generalization bounds usually involve a complexity measure that is determined by the considered theoretical framework. This limits the scope of such analyses, as other forms of capacity measures or regularization are used in practical algorithms. In this paper, we leverage the framework of disintegrated PAC-Bayesian bounds and combine it with Gibbs distributions to derive generalization bounds involving a complexity measure that can be defined by the user. Our bounds stand in probability jointly over the hypotheses and the learning sample, which allows us to tighten the complexity for a given generalization gap since it can be set to fit both the hypothesis class and the task.

## 1 INTRODUCTION

Statistical learning theory offers various theoretical frameworks to assess generalization by studying whether the empirical risk is representative of the true risk thanks to an upper bounding strategy of the generalization gap. The generalization gap is a deviation between the true risk and the empirical risk. An upper bound on this gap is generally a function of two main quantities: *(i)* the size of the training sample and *(ii)* a complexity measure that captures how prone a model is to overfitting. One potential limitation is that existing frameworks are restricted to particular complexity measures, among them the VC-dimension (Vapnik & Chervonenkis, 1971) or the Rademacher complexity (Bartlett & Mendelson, 2002) for which some generalization bounds can be derived. To the best of our knowledge, there is no generalization bound able to take into account, by construction, some arbitrary complexity measures that can serve as good proxies for the generalization gap.

In this paper, we tackle this drawback by leveraging the framework of disintegrated PAC-Bayesian bound (Theorem 2.1) to propose a novel generalization bound with arbitrary complexity measures. To do so, we make use of the Gibbs probability distributions (Equation (2)) that depend on a user-defined parametric function characterizing the complexity. It allows us to derive guarantees in terms of probabilistic bounds that depend on a model sampled from a Gibbs distribution mentioned above. It is worth noticing that our result allows retrieving the uniform convergence and algorithm-dependent bounds.

We believe that our novel result provides theoretical foundations for the many regularizations used in practice to perform model selection. For instance, our result allows integrating complexity measures studied empirically in a recent line of work on over-parametrized models (Jiang et al., 2019; Dziugaite et al., 2020; Jiang et al., 2021). In our experimental evaluation, we show how these measures can be easily integrated into our framework in practice. We notably provide a stochastic version of the Metropolis Adjusted Langevin algorithm to compute empirical estimates of our bounds.

**Organization of the paper.** In Section 2, we provide some preliminary definitions and concepts. Then, we present our main contribution in Section 3. In Section 4, we provide a practical instantiation of our framework before concluding in Section 5.

## 2 PRELIMINARIES

### 2.1 SETTING

We consider the supervised classification learning setting where $\mathbb{X}$ denotes the input space and $\mathbb{Y}$ is the label space. We consider that an example $(\mathbf{x}, y) \in \mathbb{X} \times \mathbb{Y}$ is sampled from an unknown data distribution $\mathcal{D}$ on $\mathbb{X} \times \mathbb{Y}$. A learning sample $\mathcal{S} = \{(\mathbf{x}_i, y_i)\}_{i=1}^{m}$ contains $m$ examples drawn *i.i.d.* from $\mathcal{D}$; we denote the distribution of such an $m$-sample by $\mathcal{D}^m$. Let $\mathbb{H}$ be a potentially infinite set of functions $h : \mathbb{X} \to \mathbb{Y}$, called hypotheses (or models), that associate a label from $\mathbb{Y}$ given an input from $\mathbb{X}$. Let $\mathbb{M}(\mathbb{H})$ be the set of probability densities over $\mathbb{H}$ given a reference measure (*e.g.*, the Lebesgue measure); we denote by $\mathbb{M}^*(\mathbb{H}) \subseteq \mathbb{M}(\mathbb{H})$ the set of strictly positive probability densities. Given a learning sample $\mathcal{S}$, we aim to find $h \in \mathbb{H}$ that minimizes the so-called true risk $R_{\mathcal{D}}(h) = \mathbb{P}_{(\mathbf{x}, y) \sim \mathcal{D}} I[h(\mathbf{x}) \neq y]$, where $I[a] = 1$ if $a$ is true, and 0 otherwise. In practice, as the data distribution $\mathcal{D}$ is unknown, we estimate the true risk with its empirical counterpart: the empirical risk $R_{\mathcal{S}}(h) = \frac{1}{m} \sum_{i=1}^{m} I[h(\mathbf{x}_i) \neq y_i]$. We hereafter denote the generalization gap by $\phi : [0,1]^2 \to \mathbb{R}$, which is usually defined by $\phi(R_{\mathcal{D}}(h), R_{\mathcal{S}}(h)) = |R_{\mathcal{D}}(h) - R_{\mathcal{S}}(h)|$ that quantifies how much the empirical risk is representative of the true risk.

In this paper, we leverage the PAC-Bayesian framework Shawe-Taylor & Williamson (1997); McAllester (1998); Guedj (2019); Alquier (2021) to upper-bound the generalization gap with a function that depends on an *arbitrary* measure of complexity. In PAC-Bayes, we consider an *apriori* belief on the hypotheses in $\mathbb{H}$ that is modeled by a prior distribution $\pi \in \mathbb{M}^*(\mathbb{H})$ on $\mathbb{H}$. We aim to learn, from $\mathcal{S}$ and $\pi$, a *posterior* distribution $\rho \in \mathbb{M}(\mathbb{H})$ on $\mathbb{H}$ to assign higher probability to the best hypotheses in $\mathbb{H}$ (the support of $\rho$ being included in the support of $\pi$). The classical PAC-Bayesian generalization bounds provide upper bounds in expectation over $\rho$, meaning that they bound the generalization gap expressed as $|\mathbb{E}_{h \sim \rho}[R_{\mathcal{D}}(h) - R_{\mathcal{S}}(h)]|$, and where the complexity term depends on the KL divergence between $\rho$ and $\pi$ defined as $\mathrm{KL}(\rho \| \pi) = \mathbb{E}_{h \sim \rho} \ln \frac{\rho(h)}{\pi(h)}$. This standard complexity hence captures how much the prior and the posterior distribution deviate in expectation over all the hypotheses. To incorporate custom complexities in the bounds, we follow a slightly different framework recalled below (the disintegrated PAC-Bayesian bounds) in which the expectations on $\rho$ are "disintegrated": the gap $\phi(R_{\mathcal{D}}(h), R_{\mathcal{S}}(h)) = |R_{\mathcal{D}}(h) - R_{\mathcal{S}}(h)|$ of a single $h$ sampled from $\rho$ is considered in the bounds.

### 2.2 DISINTEGRATED PAC-BAYESIAN BOUNDS

The disintegrated PAC-Bayesian bounds have been introduced by Catoni (2007, Th 1.2.7) and Blanchard & Fleuret (2007, Prop 3.1)[1]. As far as we know, despite their significance, they have been little used in the literature and received only recently renewed interest for deriving tight bounds in practice (*e.g.*, Rivasplata et al. (2020); Viallard et al. (2021)). Such bounds provide guarantees for a hypothesis $h$ sampled from a posterior distribution $\rho_{\mathcal{S}}$. They take the form of a bound that stands with high probability (at least $1 - \delta$) over the random choice of training set $\mathcal{S} \sim \mathcal{D}^m$ and hypothesis $h$. This paper mainly focuses on a particular bound, namely, the one of Rivasplata et al. (2020, Theorem 1 *(i)*) recalled below.

**Theorem 2.1** (General Disintegrated Bound of Rivasplata et al. (2020)). *For any distribution $\mathcal{D}$ on $\mathbb{X} \times \mathbb{Y}$, for any hypothesis set $\mathbb{H}$, for any distribution $\pi \in \mathbb{M}^*(\mathbb{H})$, for any measurable function $\varphi : \mathbb{H} \times (\mathbb{X} \times \mathbb{Y})^m \to \mathbb{R}$, for any $\delta \in (0, 1]$, we have*

$$\mathbb{P}_{\mathcal{S} \sim \mathcal{D}^m, h \sim \rho_{\mathcal{S}}} \left[ \varphi(h, \mathcal{S}) \leq \underbrace{\ln \left[ \frac{\rho_{\mathcal{S}}(h)}{\pi(h)} \right] + \ln \left[ \frac{1}{\delta} \mathbb{E}_{\mathcal{S}' \sim \mathcal{D}^m} \mathbb{E}_{g \sim \pi} \exp \left( \varphi(g, \mathcal{S}') \right) \right]}_{\Phi(\rho_{\mathcal{S}}, \pi, \delta)} \right] \geq 1 - \delta,$$

*where $\rho_{\mathcal{S}}$ is a posterior distribution such that $\rho_{\mathcal{S}} \in \mathbb{M}(\mathbb{H})$.*

In this case, the function $\varphi(h, \mathcal{S}) = m \, \phi(R_{\mathcal{D}}(h), R_{\mathcal{S}}(h))$ is a deviation between the true risk $R_{\mathcal{D}}(h)$ and the empirical risk $R_{\mathcal{S}}(h)$. Moreover, the function $\Phi(\rho_{\mathcal{S}}, \pi, \delta)$ is constituted of 2 terms: *(i)* the *disintegrated* KL divergence $\ln \frac{\rho_{\mathcal{S}}(h)}{\pi(h)}$ defining how much the prior and posterior distributions deviate

---

[1] Disintegrated PAC-Bayesian bounds have also been introduced as a "single-draw case" by Hellström & Durisi (2020).

for a single $h$, and *(ii)* the term $\ln\left[\frac{1}{\delta}\,\mathbb{E}_{\mathcal{S}'\sim\mathcal{D}^m}\,\mathbb{E}_{g\sim\pi}\exp\left(\varphi(g,\mathcal{S}')\right)\right]$ which is constant *w.r.t.* $h\in\mathbb{H}$ and $\mathcal{S}\in(\mathbb{X}\times\mathbb{Y})^m$ and usually upper-bounded to instantiate the bound. In the following, we refer to the whole right-hand side of the bound, $\Phi()$, as the *complexity measure* for the sake of simplicity. Note that this is in slight contrast with the standard definitions of complexity, where the term *(ii)* (related to $\delta$ and the sample size $m$) is not included. This additional term is, in fact, constant *w.r.t.* the hypothesis $h\sim\rho_\mathcal{S}$ and the learning sample $\mathcal{S}\sim\mathcal{D}^m$.

In the bound of Theorem 2.1, the complexity term $\Phi()$ depends on the disintegrated KL divergence and suffers from drawbacks: the KL complexity term is imposed by the framework and can be subject to high variance in practice (Viallard et al., 2021). However, it is important to notice that this disintegrated KL divergence has a clear advantage: it only depends on the hypothesis $h$ and data sample $\mathcal{S}$, instead of the whole hypothesis class (as it is often the case for instance with the KL divergence in PAC-Bayesian bounds, or the VC-dimension). This might imply a better correlation between the generalization gap and some complexity measures. In the next section, we leverage this disintegrated KL divergence to derive our main contribution: a general bound that involves arbitrary complexity measures.

## 3 Integrating Arbitrary Complexities in Generalization Bounds

We first begin with a short presentation of our result to give some preliminary intuitions and to introduce the notion of Gibbs distribution which is a key element in the exposition of our contribution. We then formalize our theoretical result in Section 3.3.

### 3.1 An Introduction to our Results

Let $\Phi_\mu(h,\mathcal{S},\delta)$ be a real-valued function that takes a hypothesis $h\in\mathbb{H}$, a learning sample $\mathcal{S}\in(\mathbb{X}\times\mathbb{Y})^m$, and the parameter $\delta$ as arguments and that is dependent on an additional function $\mu:\mathbb{H}\times(\mathbb{X}\times\mathbb{Y})^m\to\mathbb{R}$. The idea is to use this function $\mu()$ to parametrize the complexity measure with respect to the data sample $\mathcal{S}$ and the model $h$, in order to introduce custom complexity measures in the bound; we call "*parametric function*" the function $\mu()$. This function must, in fact, serves to obtain a complexity measure $\Phi_\mu(h,\mathcal{S},\delta)$ that is representative of the generalization gap (which is unknown). For instance, when $\mathbb{H}$ is a set of hypotheses $h_\mathbf{w}$ parameterized by some weights $\mathbf{w}\in\mathbb{R}^d$, we can fix $\mu(h_\mathbf{w},\mathcal{S})=\|\mathbf{w}\|$, for some norm $\|\cdot\|$. This means that $\mu(h_\mathbf{w},\mathcal{S})$ can be set to the regularization term of the chosen objective function so that the complexity, hence the bound, will depend on it. This is not entirely new since, for example, uniform stability bounds allow one to consider such norms (see, *e.g.*, Kakade et al., 2008). This example is just for illustration purposes. Our framework is compatible with broader families of complexity measures, as we will see later. Given such a parametric function $\mu()$, the bound we derive in Theorem 3.1 takes the following form.

**Definition 3.1** (Generalization Bound with Complexity Measures). *Let $\phi:[0,1]^2\to\mathbb{R}$ be the generalization gap, $\mu:\mathbb{H}\times(\mathbb{X}\times\mathbb{Y})^m\to\mathbb{R}$ be a parametric function. A generalization bound with arbitrary complexity measures is defined such that if for any distribution $\mathcal{D}$ on $\mathbb{X}\times\mathbb{Y}$, for any hypothesis set $\mathbb{H}$, there exists a real-valued function $\Phi_\mu:\mathbb{H}\times(\mathbb{X}\times\mathbb{Y})^m\times(0,1]\to\mathbb{R}$ such that for any $\delta\in(0,1]$, we have*

$$\mathbb{P}_{\mathcal{S}\sim\mathcal{D}^m,\,h\sim\rho_\mathcal{S}}\left[\phi(R_\mathcal{D}(h),R_\mathcal{S}(h))\leq\Phi_\mu(h,\mathcal{S},\delta)\right]\geq 1-\delta. \tag{1}$$

The main trick to obtain such a result is to consider a particular posterior distribution $\rho_\mathcal{S}$: we incorporate the function $\mu()$ by choosing the distribution $\rho_\mathcal{S}$ as the Gibbs distribution defined as

$$\rho_\mathcal{S}(h)\propto\exp\left[-\alpha R_\mathcal{S}(h)-\mu(h,\mathcal{S})\right],\quad\text{where }\alpha\in\mathbb{R}^+. \tag{2}$$

This Gibbs distribution $\rho_\mathcal{S}$ is interesting from an optimization viewpoint: a hypothesis $h$ is more likely to be sampled from it when the objective function $h\mapsto R_\mathcal{S}(h)+\frac{1}{\alpha}\mu(h,\mathcal{S})$ is low for a given $\mathcal{S}$. In the ideal case, since we want to minimize the generalization gap $\phi(R_\mathcal{D}(h),R_\mathcal{S}(h))$, one can define the function $\mu(h,\mathcal{S})=\alpha\phi(R_\mathcal{D}(h),R_\mathcal{S}(h))-\alpha R_\mathcal{S}(h)$ to obtain a Gibbs distribution that samples hypotheses with small gaps. However, since the generalization gaps are unknown, they must be replaced with a computable function $\mu()$. For instance, the function $\mu()$ can serve as a "regularizing term" (when $\mu()$ is a norm), so that a hypothesis is more likely to be sampled when the trade-off $R_\mathcal{S}(h)+\frac{1}{\alpha}\mu(h,\mathcal{S})$ is low. Equation (2) might look restrictive, but it can actually represent

any probability density function. Indeed, let $\rho'_{\mathcal{S}}$ be a distribution on $\mathbb{H}$, *e.g.*, a Gaussian or a Laplace distribution, by setting $\mu(h, \mathcal{S}) = -\alpha R_{\mathcal{S}}(h) - \ln \rho'_{\mathcal{S}}(h)$ we can retrieve the distribution $\rho'_{\mathcal{S}}$. The Gibbs distribution is well-known and studied in learning theory. In the following, we discuss the principal theoretical works based on it and highlight the differences with our framework.

## 3.2 RELATED WORKS USING THE GIBBS DISTRIBUTION

This section highlights two lines of work that are related to our setting: *(i)* the link between the Gibbs distribution and optimization and *(i)* the usage of the Gibbs distribution in generalization bounds.

**Relationship between optimization and the Gibbs distribution.** Given an objective function $f : \mathbb{H} \times (\mathbb{X} \times \mathbb{Y})^m \to \mathbb{R}$, the information risk minimization principle (Zhang, 2006) is related to the Gibbs distribution, *i.e.*, by taking

$$\rho_{\mathcal{S}} = \underset{\rho \in \mathbb{M}(\mathbb{H})}{\operatorname{argmin}} \left\{ \underset{h \sim \rho}{\mathbb{E}} f(h, \mathcal{S}) + \frac{\mathrm{KL}(\rho \| \pi)}{\alpha} \right\} \quad \text{where} \quad \rho_{\mathcal{S}}(h) \propto \exp\left[-\alpha f(h, \mathcal{S}) + \ln \pi(h)\right].$$

Note that in our case, we have $f(h, \mathcal{S}) = R_{\mathcal{S}}(h) + \frac{1}{\alpha}\mu(h, \mathcal{S}) - \frac{1}{\alpha}\ln \pi(h)$. This distribution is also linked to the Stochastic Gradient Langevin Dynamics (SGLD) algorithm (Welling & Teh, 2011) that learns the hypothesis $h \in \mathbb{H}$ by running several iterations of the form

$$h_t \longleftarrow h_{t-1} - \beta \nabla f(h, \mathcal{S}) + \sqrt{\frac{2\beta}{\alpha}} \epsilon_t, \quad \text{with} \quad \epsilon_t \sim \mathcal{N}(\mathbf{0}, \mathbf{I}_D), \tag{3}$$

where $h_t$ is the hypothesis learned at iteration $t \in \mathbb{N}$, $\beta$ is the learning rate, and $\alpha$ is the concentration parameter of the Gibbs distribution. This algorithm has an interesting feature: when the learning rate $\beta$ tends to zero, the SGLD algorithm becomes a continuous-time process called Langevin diffusion, defined as the stochastic differential equation in Equation (4). Indeed, Equation (3) can be seen as the Euler-Maruyama discretization (see, Raginsky et al., 2017) of Equation (4) defined for $t \geq 0$ as

$$dh_t = -\nabla f(h_t, \mathcal{S})dt + \sqrt{2\alpha} B_t, \tag{4}$$

where $B_t$ is the Brownian motion. Under some mild assumptions on the function $f()$, Chiang et al. (1987) show that the invariant distribution of the Langevin diffusion is the Gibbs distribution proportional to $\exp(-\alpha f(h_t, \mathcal{S}))$.

**Gibbs distributions in generalization bounds.** The Gibbs distribution is introduced in the PAC-Bayesian theory by Catoni (2004; 2007). Alquier et al. (2016, Theorems 4.2 & 4.3) further develop PAC-Bayesian generalization bounds based on the Gibbs distribution of Equation (2) with $\mu(h, \mathcal{S}) = 0$ as posterior. The Gibbs distribution has also been considered in information-theoretic generalization bounds (see *e.g.*, Xu & Raginsky, 2017; Goyal et al., 2017; Bu et al., 2020) that upper-bound the expected generalization gap $\mathbb{E}_{\mathcal{S} \sim \mathcal{D}^m, h \sim \rho_{\mathcal{S}}} R_{\mathcal{D}}(h) - R_{\mathcal{S}}(h)$. For instance, Kuzborskij et al. (Theorem 1, 2019) provides generalization bounds for $f$ being the empirical risk (with sub-Gaussian losses). Aminian et al. (Theorem 1, 2021) prove a closed-form solution of the expected generalization gap with the Gibbs distribution defined with a non-negative $f$. The expected true risk $\mathbb{E}_{\mathcal{S} \sim \mathcal{D}^m, h \sim \rho_{\mathcal{S}}} R_{\mathcal{D}}(h)$ has also been upper bounded by excess risk bounds (Xu & Raginsky, 2017; Kuzborskij et al., 2019), *i.e.*, bounds *w.r.t.* the minimal true risk over the hypothesis set. However, all these bounds consider expected risks while we are interested in the risk of a *single* hypothesis $h$ sampled from $\rho_{\mathcal{S}}$. Hence, to the best of our knowledge, we are the first to derive probabilistic bounds for a single hypothesis sampled from a Gibbs distribution (see Corollary 3.1, Theorem 3.1).

## 3.3 OUR MAIN RESULT: GENERALIZATION BOUND WITH COMPLEXITY MEASURES

We now state our main result: a bound on the generalization gap involving a custom $\mu$, standing for hypotheses sampled from the posterior $\rho_{\mathcal{S}}(h) \propto \exp\left[-\alpha R_{\mathcal{S}}(h) - \mu(h, \mathcal{S})\right]$.

**Theorem 3.1** (Generalization Bound with Complexity Measures). *Let $\phi : [0, 1]^2 \to \mathbb{R}$ be the generalization gap. For any $\mathcal{D}$ on $\mathbb{X} \times \mathbb{Y}$, for any hypothesis set $\mathbb{H}$, for any prior distribution*

$\pi \in \mathbb{M}^*(\mathbb{H})$ *on* $\mathbb{H}$, *for any* $\mu : \mathbb{H} \times (\mathbb{X} \times \mathbb{Y})^m \to \mathbb{R}$, *for any* $\delta \in (0, 1]$, *we have*

$$\mathbb{P}_{\mathcal{S} \sim \mathcal{D}^m, \, h' \sim \pi, \, h \sim \rho_{\mathcal{S}}} \left[ \phi(R_{\mathcal{D}}(h), R_{\mathcal{S}}(h)) \leq \left[ \alpha R_{\mathcal{S}}(h') + \mu(h', \mathcal{S}) \right] - \left[ \alpha R_{\mathcal{S}}(h) + \mu(h, \mathcal{S}) \right] \right.$$
$$\left. + \ln \frac{\pi(h')}{\pi(h)} + \ln \left( \frac{4}{\delta^2} \, \mathbb{E}_{\mathcal{S}' \sim \mathcal{D}^m} \, \mathbb{E}_{g \sim \pi} \exp \left[ \phi(R_{\mathcal{D}}(g), R_{\mathcal{S}'}(g)) \right] \right) \right] \geq 1 - \delta,$$

*where* $\rho_{\mathcal{S}}$ *is the Gibbs distribution defined by Equation* (2).

This theorem is general since it depends only on the functions $\phi()$ (expressing the generalization gap) and $\mu()$ (expressing the complexity) chosen by the user. Moreover, we show that this theorem allows obtaining uniform-convergence-based and algorithm-dependent bounds with the integration of complexity measures. We defer the proof of this result to Appendix D.

Given $\phi()$ and $\mu()$, we note a point that can be surprising at first reading: it appears indeed possible to sample hypotheses with a high objective $R_{\mathcal{S}}(h) + \frac{1}{\alpha} \mu(h, \mathcal{S})$ value and to obtain a tight generalization bound. However, by definition of the Gibbs distribution $\rho_{\mathcal{S}}$, such a sampled hypothesis $h \sim \rho_{\mathcal{S}}$ is less likely to be drawn since the density is higher when the objective is low. In other words, when $\mu(h, \mathcal{S})$ acts as a regularizer, the bound holds more likely for the hypotheses achieving a low regularized empirical risk, which is a rather expected result when considering regularized learning.

In general, the bound may appear loose as there is no explicit dependence on the size of the data sample $m$. However, to get a bound that converges when $m$ increases, it is sufficient to fix $\phi()$ as a function of $m$ such as $\phi(R_{\mathcal{D}}(h), R_{\mathcal{S}}(h)) = m \operatorname{kl}[R_{\mathcal{S}}(h) \| R_{\mathcal{D}}(h)]$ or $\phi(R_{\mathcal{D}}(h), R_{\mathcal{S}}(h)) = 2m[R_{\mathcal{D}}(h) - R_{\mathcal{S}}(h)]^2$ where $\operatorname{kl}(q\|p) \triangleq q \ln \frac{q}{p} + (1 - q) \ln \frac{1-q}{1-p}$ for $p \in (0, 1)$ and $q \in [0, 1]$. Then, the tightness of the bound depends on $m$, apart from $\phi()$, $\mu()$ and $\alpha$.

The remaining challenge is to upper-bound $\mathbb{E}_{\mathcal{S}' \sim \mathcal{D}^m} \mathbb{E}_{g \sim \pi} \exp[\phi(R_{\mathcal{D}}(g), R_{\mathcal{S}'}(g))]$ and $\ln \frac{\pi(h')}{\pi(h)}$ to get a practical bound. As an illustration, we provide in the next corollary an instantiation of Theorem 3.1 for two generalization gaps: $\phi(R_{\mathcal{D}}(h), R_{\mathcal{S}}(h)) = m \operatorname{kl}[R_{\mathcal{S}}(h) \| R_{\mathcal{D}}(h)]$ and $\phi(R_{\mathcal{D}}(h), R_{\mathcal{S}}(h)) = 2m[R_{\mathcal{D}}(h) - R_{\mathcal{S}}(h)]^2$; and for $\pi$ is a uniform distribution on a bounded set $\mathbb{H}$.

**Corollary 3.1** (Practical Generalization Bound with Complexity Measures). *For any* $\mathcal{D}$ *on* $\mathbb{X} \times \mathbb{Y}$, *for any bounded hypothesis set* $\mathbb{H}$, *given the uniform prior* $\pi$ *on* $\mathbb{H}$, *for any* $\mu : \mathbb{H} \times (\mathbb{X} \times \mathbb{Y})^m \to \mathbb{R}$, *for any* $\delta \in (0, 1]$, *with probability at least* $1 - \delta$ *over* $\mathcal{S} \sim \mathcal{D}^m$, $h' \sim \pi$, $h \sim \rho_{\mathcal{S}}$ *we have*

$$\operatorname{kl}\left[R_{\mathcal{S}}(h) \| R_{\mathcal{D}}(h)\right] \leq \frac{1}{m} \left[ \left[ \alpha R_{\mathcal{S}}(h') + \mu(h', \mathcal{S}) \right] - \left[ \alpha R_{\mathcal{S}}(h) + \mu(h, \mathcal{S}) \right] + \frac{8\sqrt{m}}{\delta^2} \right]_+, \quad (5)$$

*and* $\left| R_{\mathcal{D}}(h) - R_{\mathcal{S}}(h) \right| \leq \sqrt{\frac{1}{2m} \left[ \left[ \alpha R_{\mathcal{S}}(h') + \mu(h', \mathcal{S}) \right] - \left[ \alpha R_{\mathcal{S}}(h) + \mu(h, \mathcal{S}) \right] + \frac{8\sqrt{m}}{\delta^2} \right]_+}, \quad (6)$

*where* $[a]_+ = \max(0, a)$, *and* $\rho_{\mathcal{S}}$ *is the Gibbs distribution defined in Equation* (2).

Interestingly, Corollary 3.1 gives a bound on $\operatorname{kl}[R_{\mathcal{S}}(h) \| R_{\mathcal{D}}(h)]$ and $|R_{\mathcal{D}}(h) - R_{\mathcal{S}}(h)|$ where all terms except $R_{\mathcal{D}}(h)$ are computable. To compute Equations (5) and (6) we can rearrange the terms to obtain a generalization bound on the true risk $R_{\mathcal{D}}(h)$. We obtain respectively

$$R_{\mathcal{D}}(h) \leq \overline{\operatorname{kl}} \left( R_{\mathcal{S}}(h) \, \middle| \, \frac{1}{m} \left[ [\alpha R_{\mathcal{S}}(h') + \mu(h', \mathcal{S})] - [\alpha R_{\mathcal{S}}(h) + \mu(h, \mathcal{S})] + \frac{8\sqrt{m}}{\delta^2} \right]_+ \right), \quad (7)$$

$$\text{and } R_{\mathcal{D}}(h) \leq R_{\mathcal{S}}(h) + \sqrt{\frac{1}{2m} \left[ [\alpha R_{\mathcal{S}}(h') + \mu(h', \mathcal{S})] - [\alpha R_{\mathcal{S}}(h) + \mu(h, \mathcal{S})] + \frac{8\sqrt{m}}{\delta^2} \right]_+}, \quad (8)$$

where $\overline{\operatorname{kl}}(q|\tau) = \max\{p \in (0, 1) \mid \operatorname{kl}(q\|p) \leq \tau\}$. These bounds are used in Section 4 to illustrate the generalization guarantees for different values of $\mu()$ and $\alpha$. In general, Equation (7) provides a tighter bound on the true risk than Equation (8). This can be proven with Pinsker's inequality (Appendix G) and is shown in our experiments. Notice that the r.h.s. of Equations (5) and (6) enjoys asymptotic convergence for $m \to \infty$. However, for some trivial cases, the convergence rate can be arbitrarily degraded by increasing $[\alpha R_{\mathcal{S}}(h') + \mu(h', \mathcal{S})] - [\alpha R_{\mathcal{S}}(h) + \mu(h, \mathcal{S})]$. For example,

for a large empirical risk $R_\mathcal{S}(h')$ (which is common when $h'$ is sampled from a uniform prior on $\mathbb{H}$), and for $\alpha{=}m$ and $\mu(h,\mathcal{S}){=}0$, the r.h.s. for $\phi(R_\mathcal{D}(h),R_\mathcal{S}(h)) = \mathrm{kl}[R_\mathcal{S}(h)\|R_\mathcal{D}(h)]$ simplifies to $\Phi_\mu(h,\mathcal{S},\delta) = [[R_\mathcal{S}(h'){-}R_\mathcal{S}(h)] + \frac{1}{m}\ln\frac{2\sqrt{m}}{\delta}]_+$ and is large, no matter $m$. In order for the bound to be meaningful, we have then to set $\alpha$ and $\mu()$ such that *(i)* the distribution $\rho_\mathcal{S}$ allows us to sample a hypothesis $h$ associated with a low objective function $h \mapsto R_\mathcal{S}(h)+\frac{1}{\alpha}\mu(h,\mathcal{S})$ and *(ii)* the complexity measure $\Phi_\mu(h,\mathcal{S},\delta)$ is tight. For example, for $\alpha{=}\sqrt{m}$ and $\mu(h,\mathcal{S}){=}0$, the distribution $\rho_\mathcal{S}$ is less concentrated around the minimizers of the empirical risk, but the complexity measure is tighter compared to the previous example: $[\frac{1}{\sqrt{m}}[R_\mathcal{S}(h'){-}R_\mathcal{S}(h)] + \frac{1}{m}\ln\frac{2\sqrt{m}}{\delta}]_+$. Lastly, in the ideal case with $\mu(h,\mathcal{S}){=}\frac{\alpha}{m}\phi(R_\mathcal{D}(h),R_\mathcal{S}(h)){-}\alpha R_\mathcal{S}(h)$ and $\alpha{=}\sqrt{m}$, the upper-bound of $\phi(R_\mathcal{D}(h),R_\mathcal{S}(h)) = m\,\mathrm{kl}[R_\mathcal{S}(h)\|R_\mathcal{D}(h)]$ becomes $[\frac{1}{\sqrt{m}}(\mathrm{kl}[R_\mathcal{S}(h')\|R_\mathcal{D}(h')]{-}\mathrm{kl}[R_\mathcal{S}(h)\|R_\mathcal{D}(h)]) + \frac{1}{m}\ln\frac{2\sqrt{m}}{\delta}]_+$ which is tight when the gaps of $h$ and $h'$ are small; the tightness arise with high probability since the density $\rho_\mathcal{S}(h)\propto\exp(-\frac{\alpha}{m}\phi(R_\mathcal{D}(h),R_\mathcal{S}(h)))$ is concentrated around the small gaps. This also highlights that the choice of the parametric function $\mu()$ is key to obtaining a tight generalization bound.

In our previous analysis, we considered a uniform distribution for the prior $\pi$ for illustration purposes. It is nevertheless clear that if the prior is good, *i.e.*, it associates a higher probability to hypotheses having a low objective function, then the bounds become tighter. The most favorable case is when both the prior $\pi$ and the posterior $\rho_\mathcal{S}$ associate high probabilities to these hypotheses. While the posterior $\rho_\mathcal{S}$ is generally learned from data, the choice of the prior $\pi$ matters to get tight bounds. When no prior knowledge of the problem is available, to obtain better bounds, one solution is to consider data-dependent priors that have been heavily used in the PAC-Bayesian literature (see, *e.g.*, Parrado-Hernández et al., 2012; Dziugaite et al., 2021; Pérez-Ortiz et al., 2021). In the context of our practical evaluation hereafter, we consider only uniform distributions for the prior $\pi$, as we think it helps us assess generalization better. Indeed, a hypothesis $h$ sampled from the uniform distribution $\pi$ has a high chance of underfitting. Hence, if the hypothesis $h \sim \rho_\mathcal{S}$ has a tight bound, it must be that this hypothesis generalizes well. On the other hand, when using data-dependent priors, we cannot tell if the bound is tight because the hypothesis generalizes well or because the posterior is close to the prior.

## 4 USING ARBITRARY COMPLEXITIES IN PRACTICE

The bound of Corollary 3.1 is not directly applicable in practice: the remaining challenge is to sample $h$ from the Gibbs distribution $\rho_\mathcal{S}$ defined in Equation (2). We address the sampling issue in Section 4.1. Then, we make use of the proposed solution to assess our bound in practice. Section 4.2 introduces our experimental setting and Section 4.3 reports an overview of results on the tightness of the bound. We report more results on the influence of $\alpha$ and the other parameters in Appendix E.

### 4.1 SAMPLING FROM THE GIBBS DISTRIBUTION

Sampling from the Gibbs distribution of Equation (2) is a hard task: naively, it requires to evaluate the function $h \mapsto -\alpha R_\mathcal{S}(h){-}\mu(h,\mathcal{S})$ for all $h \in \mathbb{H}$, which is intractable when $\mathbb{H}$ is infinite or even large. In an empirical study of our bound, we tackle this issue for over-parameterized models, which we later consider in Section 4.2. Let us consider a set $\mathbb{H}$ of hypotheses $h_\mathbf{w}$ parameterized by $\mathbf{w} \in \mathbb{R}^D$, and a tractable distribution denoted $P_\mathcal{U}^\mathbf{w}$ (*e.g.*, a Gaussian distribution) such that its density approximates the density of $\rho_\mathcal{S}$. In this setting, to learn such a tractable distribution, we propose in Algorithm 1 a stochastic version of the Metropolis Adjusted Langevin Algorithm (MALA, Besag (1994))[2]. Its objective is to generate samples from $\rho_\mathcal{S}$ by iteratively refining the tractable distribution that we define as

$$P_\mathcal{U}^\mathbf{w} = \mathcal{N}\left(\mathbf{w}{-}\beta\nabla\left[R_\mathcal{U}^\ell(\mathbf{w})+\frac{1}{\alpha}\mu(\mathbf{w},\mathcal{U})\right], \frac{2\beta}{\alpha}\mathbf{I}\right), \tag{9}$$

where $R_\mathcal{U}^\ell(\mathbf{w}) = \mathbb{E}_{(\mathbf{x},y)\sim\mathcal{U}}\,\ell(h_\mathbf{w},(\mathbf{x},y))$ is the empirical risk on the mini-batch $\mathcal{U} \subseteq \mathcal{S}$, and $\ell : \mathbb{H} \times (\mathbb{X}{\times}\mathbb{Y}) \to [0,1]$ is a loss function. Concretely, we initialize the parameters $\mathbf{w}$ of the model as the output of an optimization algorithm (Vanilla SGD in our case) minimizing $R_\mathcal{S}^\ell(\mathbf{w})+\frac{1}{\alpha}\mu(\mathbf{w},\mathcal{S})$ (which is approximated by $R_\mathcal{U}^\ell(\mathbf{w})+\frac{1}{\alpha}\mu(\mathbf{w},\mathcal{U})$ for each mini-batch $\mathcal{U}$).

---

[2]See Chib & Greenberg (1995) for an introduction on Metropolis-Hastings Algo on which MALA is based.

---

**Algorithm 1** Stochastic MALA

1: **Input:** Learning set $\mathcal{S}$, weights $\mathbf{w}$, function $\mu()$, loss function $\ell()$
2: **Hyperparameters:** Number of iterations $T$, learning rate $\beta$, parameter $\alpha$
3: **for** $t \leftarrow 1 \ldots T$ **do**
4:     $\mathcal{U} \leftarrow$ Sample (without replacement) a mini-batch from $\mathcal{S}$
5:     $\mathbf{w}' \leftarrow$ Sample from the distribution $\mathrm{P}_{\mathcal{U}}^{\mathbf{w}}$
6:     $\tau \leftarrow \min\left(1, \frac{\rho_{\mathcal{U}}(\mathbf{w}')\mathrm{P}_{\mathcal{U}}^{\mathbf{w}'}(\mathbf{w})}{\rho_{\mathcal{U}}(\mathbf{w})\mathrm{P}_{\mathcal{U}}^{\mathbf{w}}(\mathbf{w}')}\right)$
7:     $u \leftarrow$ Sample from the distribution $\mathrm{Uni}(0,1)$
8:     **if** $u \leq \tau$ **then**
9:         $\mathbf{w} \leftarrow \mathbf{w}'$
10: **return** $\mathbf{w}$

---

Then, we refine them as follows: at each iteration, given the current weights $\mathbf{w}$ and a mini-batch $\mathcal{U} \subseteq \mathcal{S}$ (Line 4), we sample a candidate vector $\mathbf{w}'$ (Line 5) according to the distribution $\mathrm{P}_{\mathcal{U}}^{\mathbf{w}}$; then (Line 6 to 9) we decide to reject or accept the new candidate to become our current weights $\mathbf{w}$, depending on its ratio $\tau = \min\left(1, \frac{\rho_{\mathcal{U}}(\mathbf{w}')\mathrm{P}_{\mathcal{U}}^{\mathbf{w}'}(\mathbf{w})}{\rho_{\mathcal{U}}(\mathbf{w})\mathrm{P}_{\mathcal{U}}^{\mathbf{w}}(\mathbf{w}')}\right)$ is larger than a control value $u$ sampled from the uniform distribution $\mathrm{Uni}(0,1)$ on $[0,1]$. Note that, to compute $\tau$, it is not necessary to know the normalization constants of the two distributions appearing in $\tau$ since they cancel out. In other words, only the function (without the normalizations) associated to the distributions are required to compute $\tau$. Under the mild assumption that $\rho_{\mathcal{S}}$ is absolute continuous *w.r.t.* $\mathrm{P}_{\mathcal{S}}^{\mathbf{w}}$ (see Chib & Greenberg, 1995, for details), when the number of iterations tends to infinity and when $\mathcal{U} = \mathcal{S}$, the returned $\mathbf{w}$ is sampled according to $\rho_{\mathcal{S}}$ (Smith & Roberts, 1993). Note that this assumption requires that the tractable distribution $\mathrm{P}_{\mathcal{S}}^{\mathbf{w}}$ has a strictly positive density when the density of $\rho_{\mathcal{S}}$ is strictly positive as well (see Chib & Greenberg, 1995).

### 4.2 EXPERIMENTAL SETTING

In this section, we investigate the tightness of our bounds of Equations (7) and (8) on the MNIST (Le-Cun et al., 1998) and FashionMNIST (Xiao et al., 2017) datasets. We keep the original learning set as $\mathcal{S}$ and the original test set $\mathcal{T}$ to estimate the true risk that we refer to as test risk $\mathrm{R}_{\mathcal{T}}(h)$.

**Model.** We use a "Convolutional Network in Network" (Lin et al., 2013) similarly to Jiang et al. (2019) and Dziugaite et al. (2020) , that consists of several modules of 3 convolutional layers each followed by a leaky ReLU activation function (its negative slope is set to $10^{-2}$). The depth of the network $L$ is the number of convolutional layers, and the width $H$ is the number of channels of each convolution. In addition, for each layer $i$, we denote its weights by $\mathbf{w}_i$. For full details of the architecture, we refer the reader to Appendix E. We consider $L \in \{9, 12, 15\}$ and $H \in \{128, 256\}$. Furthermore, we initialize the network with the weights $\mathbf{w}^0 \in \mathbb{R}^D$ obtained by the uniform Kaiming He initializer He et al. (2015). The set $\mathbb{H}$ corresponds to the hypotheses $h_{\mathbf{w}}$ that can be obtained from this initialization (and we clamp the weights during the optimization in the initialization interval).

**Arbitrary complexity measures.** We study 6 different complexity measures parametrized by different functions $\mu(h_{\mathbf{w}}, \mathcal{S})$ from Jiang et al. (2019, Sec. C)[3]. These 6 functions are actually independent of the learning sample $\mathcal{S}$ ($\mathcal{S}$ is dropped below for convenience) and defined as follows:

$$\mathrm{DIST\_FRO}(h_{\mathbf{w}}) = \sum_{i=1}^{L} \|\mathbf{w}_i - \mathbf{w}_i^0\|_2, \quad \text{and} \quad \mathrm{DIST\_L2}(h_{\mathbf{w}}) = \|\mathbf{w} - \mathbf{w}^0\|_2,$$

$$\text{and} \quad \mathrm{PARAM\_NORM}(h_{\mathbf{w}}) = \sum_{i=1}^{L} \|\mathbf{w}_i\|_2^2, \quad \text{and} \quad \mathrm{PATH\_NORM}(h_{\mathbf{w}}) = \sum_{i=1}^{\mathrm{card}(\mathbb{Y})} h_{\mathbf{w}^2}(\mathbf{1})[i],$$

$$\text{and} \quad \mathrm{SUM\_FRO}(h_{\mathbf{w}}) = L\left(\prod_{i=1}^{L} \|\mathbf{w}_i\|_2^2\right)^{\frac{1}{L}}, \quad \text{and} \quad \mathrm{ZERO}(h_{\mathbf{w}}) = 0.$$

---

[3] Note we consider a subset of the functions studied by Jiang et al.: we select those that are optimizable.

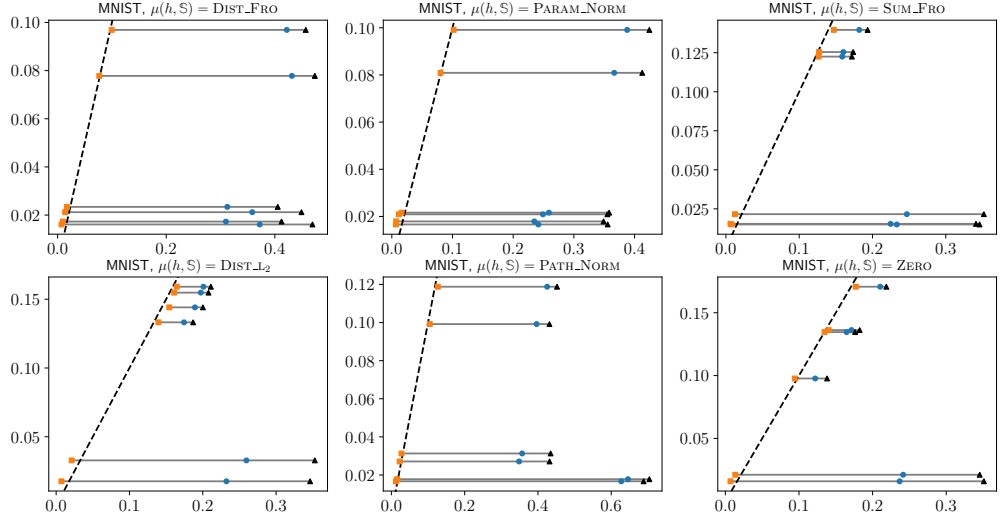

Figure 1: Scatter plot given a parametric function $\mu(h, \mathcal{S})$, where each segment represents a neural network $h_{\mathbf{w}}$ learned with a given $\alpha$, width $H$ and depth $L$. Each segment has a corresponding orange square and a blue circle. The orange squares corresponds to the empirical risk $R_{\mathcal{S}}(h)$ (x-axis) and test risk $R_{\mathcal{T}}(h)$ (y-axis). The blue circle *resp.* the black triangle represents Equation (7) *resp.* Equation (8) in the x-axis and the test risk $R_{\mathcal{T}}(h)$ in the y-axis. The dashed line is the identity function.

We define the considered measures with $\alpha$ taken among 5 values uniformly spaced between $[\sqrt{m}, m]$. Note that, as mentioned above, these 6 parametric functions are independent of the sample $\mathcal{S}$, we have also analyzed other parametric functions that depend on $\mathcal{S}$. The results obtained are similar, we decided to defer these results in Appendix E.

**Bound optimization.** To compute our bound in Equations (7) and (8), we aim to sample a hypothesis $h \sim \rho_{\mathcal{S}}$ via Algorithm 1. We set the loss function to the bounded cross entropy from Dziugaite & Roy (2018): $\ell(h, (\mathbf{x}, y)) = -\frac{1}{4} \ln(e^{-4} + (1 - 2e^{-4})h[y])$, where $h[y]$ is the probability assigned to label $y$ by $h$. The advantage of Dziugaite & Roy (2018)'s cross-entropy is that it lies in $\ell(h, (\mathbf{x}, y)) \in [0, 1]$, whereas the classical cross-entropy is unbounded. Indeed, taking into account the classical cross-entropy when optimizing the objective function would lead to focusing too much on the risk minimization, while we want to take into account $\frac{1}{\alpha} \mu(\mathbf{w}, \mathcal{U})$. We initialize the weights $\mathbf{w} \in \mathbb{R}^D$ to the solution found by optimizing the objective function $R_{\mathcal{S}}^{\ell}(\mathbf{w}) + \frac{1}{\alpha} \mu(\mathbf{w}, \mathcal{S})$ with a Vanilla SGD (with 10 epochs, a learning rate of $10^{-1}$, and a batch size of 64). Given these initial parameters $\mathbf{w}$, we execute Algorithm 1 for 1 epoch with a mini-batch of size 64, where $\beta = 10^{-4}$.

### 4.3 TIGHTNESS OF THE BOUNDS

For each parametric function $\mu()$, we report in Figures 1 and 2, the test risks $R_{\mathcal{T}}(h)$ and the values of the tightest bound on $R_{\mathcal{D}}(h)$ (*w.r.t.* $\alpha$) associated to Equations (7) and (8) for different parameters (depth $L$, width $H$). First of all, we observe that the bounds correctly upper-bound the test risks and that some measures lead to tighter bounds, such as SUM_FRO or DIST_L$_2$. We also remark that certain empirical risks are high, in particular, for these latter measures. This is due to the sampling of the hypothesis $h$ from the distribution $\rho_{\mathcal{S}}$: the hypothesis does not necessarily minimizes the objective function $h \mapsto R_{\mathcal{S}}(h) + \frac{1}{\alpha} \mu(h, \mathcal{S})$. We nevertheless observe that the bounds' values are higher when the empirical risk $R_{\mathcal{S}}(h)$ is low. This can be explained by the fact that $[\alpha R_{\mathcal{S}}(h') + \mu(h', \mathcal{S})] - [\alpha R_{\mathcal{S}}(h) + \mu(h, \mathcal{S})]$ is large in this case due notably to the non-informative prior $\pi$. Interestingly, when the empirical risks are a bit worse or close to the true risks, the bounds become tighter for certain parametric functions such as DIST_L$_2$ and SUM_FRO, which then appear to capture more information on the generalization capabilities. Indeed, the more the objective function $R_{\mathcal{S}}(h) + \frac{1}{\alpha} \mu(h, \mathcal{S})$ is representative of the gap of $h$, the tighter the bound. On the other hand, we can also note that for some measures such as DIST_FRO, PARAM_NORM, and ZERO (mainly for FashionMNIST), the bounds remain similar whatever the hypothesis which illustrates that these

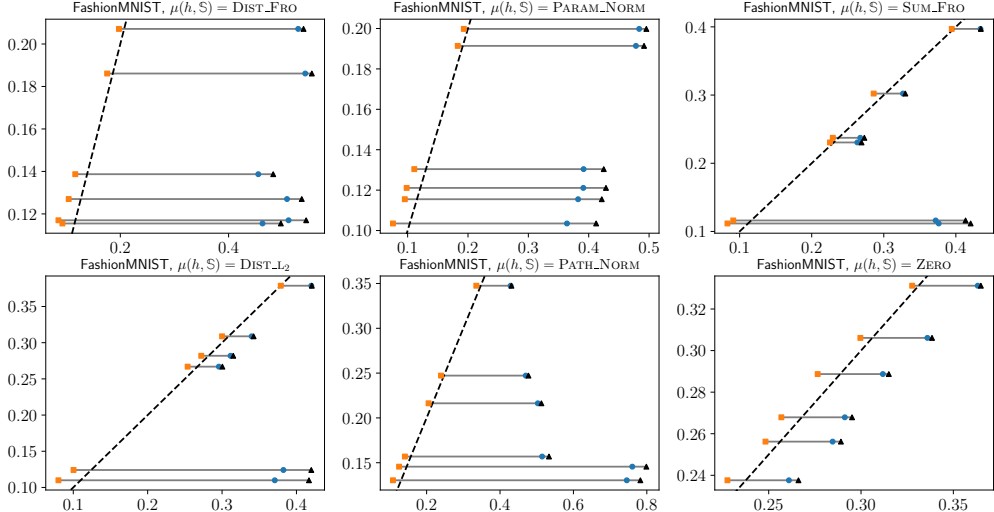

Figure 2: Scatter plot given a parametric function $\mu(h, \mathcal{S})$, where each segment represents a neural network $h_{\mathbf{w}}$ learned with a given $\alpha$, width $H$ and depth $L$. Each segment has a corresponding orange square and a blue circle. The orange squares correspond to the empirical risk $R_{\mathcal{S}}(h)$ (x-axis) and test risk $R_{\mathcal{T}}(h)$ (y-axis). The blue circle *resp.* the black triangle represents Equation (7) *resp.* Equation (8) in the x-axis and the test risk $R_{\mathcal{T}}(h)$ in the y-axis. The dashed line is the identity function.

latter measures do not really help to capture some information about the generalization gap. This confirms that there is an interest in using a parametric function that captures information on the model during the training phase to assess its generalization capability. In Appendix E, we provide additional results on the influence of the parameter $\alpha$ and the depth/width of the network. As expected, the bounds tend to increase when $\alpha$ becomes large for smaller $\alpha$ (*e.g.*, close to $\sqrt{m}$), the bounds are improved but to the price of potentially higher risks. In contrast, about the depth/width impact, some measures are less sensitive to the increase of such parameters, such as PARAM_NORM and, to a lesser extend, SUM_FRO and DIST_L$_2$. This illustrates our framework's interest in studying the impact of some regularization when learning (over-)parameterized models.

## 5 CONCLUSION

In this paper, we provide a novel generalization bound that is able to incorporate arbitrary complexity measures, unlike classical learning theory frameworks (for which the framework imposes the complexity). These measures incorporate a data and model-dependent function, which can favor tightening the complexity for the generalization gap. To the best of our knowledge, our framework is one of the few able to be general enough to bring theoretical guarantees for most of the arbitrary complexity measures used in practice, *e.g.*, based on some norms or a validation set. Such a framework may be adapted to other settings, such as transfer learning, offering new research directions. However, one limitation of this work is clearly that the hypothesis is obtained from a distribution difficult to use, namely, the Gibbs distribution, which uses a specific sampling algorithm, *e.g.*, our algorithm stochastic MALA. It would be interesting to study the performance of such a sampling theoretically. Alternately, the generality of this framework allows one to avoid the sampling if we consider uniform-convergence-type bounds, for example, as in Corollary D.1. Improving the framework in this direction is an interesting future work. In particular, investigating the use of other distributions for sampling the hypothesis could be a possible direction. Another one could be to consider other specific ways to define informative data-dependent priors in order to obtain better bounds. For instance, the parametric function can be leveraged in order to include informative prior. Another interesting perspective is to study SGD-based algorithms, either by analyzing models learned by SGD through our framework or by developing SGD alternatives to optimize our bounds. In conclusion, we believe that this work paves the way for new research directions that try to bridge statistical learning theory and practice.

## REPRODUCIBILITY STATEMENT

In order to ensure the reproducibility of our results, we complete the presentation of the experimental setup of the main text in Section 4.2 with a more complete description of the setting, models, and parameter used in Appendix E where some additional results are also provided. We also include the code of our method as an additional zip file in the supplementary material in order to facilitate the reproduction of the experiments.

Regarding the theoretical contributions, we provide in Appendices A to C the proofs of the results presented in the main paper, namely Theorems 2.1 and 3.1 and corollary 3.1. We also provide in Appendix D some additional results and discussion about the comparison of our framework with the uniform convergence and algorithm-dependent generalization bounds.

## ETHIC STATEMENT

The contributions of this paper are essentially fundamental and theoretical; we do not see an immediate potential negative social impact from these contributions. We followed classic ethical guidelines in machine learning which in our case mainly consists in bringing information about reproducibility issues which is addressed in the previous paragraph.

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
