# OpenReview forum: "Generalization Bounds with Arbitrary Complexity Measures"
_ICLR.cc/2023/Conference — Submitted to ICLR 2023_

### Official Review · Reviewer_g1xt · 2022-10-23

**Confidence:** 4
**Correctness:** 4
**Technical Novelty And Significance:** 2
**Empirical Novelty And Significance:** 2
**Recommendation:** 5

**Clarity, Quality, Novelty And Reproducibility:**

In general, I found this paper really well-written, and all the results are clear.

**Strength And Weaknesses:**

The paper is very-well written, and the idea is presented very clearly.

I think there is a mismatch between the motivation explained in the intro and the results: In particular, in the intro there is a discussion on the challenges behind proving generalization of "SGD". The results mentioned in the intro state that proving generalization is difficult because most of the theoretical results for "SGD" empirically do not correlate with the generalization gap.

Later in the paper the authors provide an abstract learning algorithm based on "Gibbs posterior" which takes into account the complexity measure. Also, there are some technical details such as, the loss function is not cross entropy,....

Another point to consider is that in the learning theory community, the main question is why does SGD without any "regularization" generalize? It is quite different from the results in the paper since the proposed algorithm is not SGD, also the proposed algorithm is based on using an "explicit" form of regularization. Again you can contrast it with the implicit bias results for SGD.

Questions:

1- For a given problem what is the best complexity measure? Is there any theoretical insight on that?

2- Generalization can't show the learnability. I would suggest that the authors compare the performance of their algorithm with SGD in terms of the "population error"?

3- Do the results suggest that we should use the proposed algorithm compared to SGD? I agree that the proposed algorithm comes with theoretical guarantees.

4- You can't sample perfectly from the Gibbs distribution. It is interesting to study the impact of the stochastic mala on the generalization. There are a few results in the literature on the finite time generalization performance of such a samplers:

[1] Negrea, Jeffrey, et al. "Information-theoretic generalization bounds for SGLD via data-dependent estimates." Advances in Neural Information Processing Systems 32 (2019).

[2] Mou, Wenlong, et al. "Generalization bounds of sgld for non-convex learning: Two theoretical viewpoints." Conference on Learning Theory. PMLR, 2018.

**Summary Of The Paper:**

This paper considers the problem of providing generalization bounds using the disintegrated PAC-Bayesian framework. The main idea behind the paper is that we can start with an arbitrary complexity measure which provides a hierarchy over the hypotheses. Then the authors show that a particular learning algorithm based on the Gibbs posterior can be used for learning. Moreover, using the PAC-Bayesian framework we can show that the complexity measure arises in the upper bound on the generalization.

**Summary Of The Review:**

This paper is interesting. However, there is a big gap between the motivation behind the paper and the actual results. It is my main concern.

---

> ### Author Response · Authors · 2022-11-19
> **Answer to Reviewer g1xt**
>
> Thank you for your review and feedback.
>
> *About the mismatch between the motivation and the results.* We agree with the reviewer that the abstract and the introduction might be misleading for the reader. Indeed, our work is not about proving generalization bounds for models that have been learned by stochastic gradient descent. Indeed, recall that we aim to provide generalization bounds with user-defined complexity measures to overcome a major drawback in the literature: the bounds usually depend on complexity measures imposed by the framework (e.g., the VC dimension or the Rademacher Complexity). Thus, we modified these two parts of the paper in order to clarify our contribution; we put in red the sentences that we modified to ease the reading.
>
> *Question 1.* We believe that this is not an easy question and must be explored in the future. Indeed, a good measure might change depending on the hypothesis set and the learning task. However, one extreme but good theoretical example of a complexity measure is when $\mu(h,{\cal S})=\frac{\alpha}{m}\phi(R\_{\cal D}(h), R\_{\cal S}(h))-\alpha R\_{\cal S}(h)$ or in other words when we directly integrate the generalization gap in the complexity measure (which is intractable). Indeed, with this complexity measure, the bound will be tight when the sampled hypotheses $h\sim\rho\_{\cal S}$ and $h'\sim\pi$ generalize well.
>
> *Question 2.* We agree that this is interesting, but we propose to mention it as a future work since our main objective is to present our general framework for integrating arbitrary complexity measures.
>
> *Question 3.* The bound can be valid when we want to analyze the performance of a model even if the algorithm used is not stochastic MALA. Indeed, the density of the Gibbs distribution associated with a model obtained by SGD will be high if we minimize the objective function $R^{\ell}\_{\cal S}(h)+\frac{1}{\alpha}\mu(h, {\cal S})$ (where $\ell$ is, in this case, the bounded cross-entropy) and thus the bounds have a high probability of holding. Our algorithm aims to sample from the Gibbs distribution, which has the advantage of sampling a model associated with a bound that holds with high probability. Our results do not suggest that our algorithm must be used necessarily instead of SGD. Comparing SGD is an interesting perspective that seems to us to be outside the scope of the paper, which aims primarily to introduce the framework for incorporating complexity measures.
>
> *Question 4.* Indeed, it would be interesting to study as future work the performance of the sampler; we add the suggestion in conclusion.

---

### Official Review · Reviewer_H8Gk · 2022-10-23

**Confidence:** 4
**Correctness:** 3
**Technical Novelty And Significance:** 3
**Empirical Novelty And Significance:** 2
**Recommendation:** 5

**Clarity, Quality, Novelty And Reproducibility:**

This paper is novel and Reproducible, as proofs and codes are provided.

However, there are a lot of details that need to be clarified.

I do not fully understand the training process. To my understanding, you first train different models with SGD using the bounded cross-entropy loss from Dziugaite & Roy (2018). Then, using the output of SGD as initialization in Algorithm 1, a new $w$ is sampled from the Gibbs algorithm with different complexity measures $\mu(h_w,S)$. What is the loss function used in Algorithm 1 (or eq(9))? The zero-one loss or the bounded cross-entropy loss? It is mentioned in the paper that $\rho_S(w)$ (Gibbs posterior distribution) is hard to compute, then how do you compute $\rho_U(w)$ in Line 6 of Algorithm 1?

Eventually, all different complexity measures will induce a different Gibbs algorithm, so what is the point of the comparison in Figures 1 and 2? Dist L2 and Sum Fro are better regularizers in training compared to the others? Or the proposed bounds are tighter for these complexity measures? The authors are encouraged to elaborate on the insights readers can obtain from these experiments.


**Strength And Weaknesses:**

Strength:
The theoretical part of the paper is very clear. Although Thm 3.1 highly depends on the structure of the Gibbs algorithm, the resulting form is correct and novel.

Weaknesses:
One major drawback is the tightness of the bound. From Figures 1 and 2, it seems that the proposed bounds are loose in most cases and are only informative when the empirical risks are high, which is not the current operating regime for deep learning.

A lot of important references are missing. This paper completely ignores a recent line of work on information-theoretic generalization bounds. Though the original forms of these bounds are in expectation [1,2], more recent results [3] can provide high-probability generalization bounds and should be mentioned in the discussion.

[1] Xu, Aolin, and Maxim Raginsky. "Information-theoretic analysis of generalization capability of learning algorithms." Advances in Neural Information Processing Systems 30 (2017).

[2] Bu, Yuheng, Shaofeng Zou, and Venugopal V. Veeravalli. "Tightening mutual information-based bounds on generalization error." IEEE Journal on Selected Areas in Information Theory 1, no. 1 (2020): 121-130.

[3] Hellström, Fredrik, and Giuseppe Durisi. "Generalization bounds via information density and conditional information density." IEEE Journal on Selected Areas in Information Theory 1, no. 3 (2020): 824-839.

More specifically, the following two papers on the generalization error for the Gibbs algorithm that provides 1/m in expectation bounds are not discussed.

[4] Kuzborskij, Ilja, Nicolò Cesa-Bianchi, and Csaba Szepesvári. "Distribution-dependent analysis of Gibbs-ERM principle." In Conference on Learning Theory, pp. 2028-2054. PMLR, 2019.

[5] Aminian, Gholamali, Yuheng Bu, Laura Toni, Miguel Rodrigues, and Gregory Wornell. "An exact characterization of the generalization error for the Gibbs algorithm." Advances in Neural Information Processing Systems 34 (2021): 8106-8118.


**Summary Of The Paper:**

This paper leverages the framework of disintegrated PAC-Bayes bounds to derive a generalization bound for the Gibbs algorithm that involves an arbitrary complexity measure. The proposed bound stands in probability jointly over the hypotheses and the learning sample, which can be used to tighten the complexity for a given generalization gap since it can be set to fit both the hypothesis class and the task.

**Summary Of The Review:**

Overall, it is an interesting paper with some novel results. However, I feel that the authors oversell their contribution in the title and abstract. The proposed results only work for Gibbs distribution. If I understand correctly, different complexity measures $\mu(h,S)$ will induce different learning algorithms, which are hard to implement and compare in practice. Also, I would expect that the structure of the Gibbs distribution could provide a tighter bound or better converge rate, but the resulting bounds are not as tight as I mentioned above.

---

> ### Author Response · Authors · 2022-11-19
> **Answer to Reviewer H8Gk**
>
> *About overselling the paper.* Thank you for your feedback on our formulation. Our intention was not indeed to oversell the paper. Therefore, we decided to adjust the abstract and the introduction to better introduce the contribution by clarifying the context related to the Gibbs distribution. Following your remark, we also decided not to mention the empirical study of over-parameterized models as it is not a direct objective of the contribution of this paper.
>
> *About the experiments.* We have added in Section 4.1 which objective function we consider in the SGD algorithm (which is essentially the same as in stochastic MALA). Note that in Line 6 of stochastic MALA, only the normalization constants are not required, which is enough to compute $\tau$; we gave some precisions on how the threshold $\tau$ (Algo 1, ligne 6) is computed. Hence the loss associated with the objective function in stochastic MALA and SGD is the bounded cross-entropy. We also believe that the point *About the tightness.* (with a discussion added in the paper) below will also help clarify the intuitions behind the experiments.  About Figures 1 and 2, each complexity measure induces a Gibbs distribution which leads to different models and, thus, different generalization bounds. The objective of the experiments is to study the informativeness of the bounds. Notably, SUM_FRO and DIST_L2 bring better guarantees in the sense that the bounds are closer to the empirical risks.
>
> *About the tightness.* We agree that the bounds can be loose. Note that the tightness depends on mainly two factors: (1) the prior distribution and (2) the user-defined complexity measure. Concerning (1), it is well known in PAC-Bayesian theory that the prior distribution can be data-dependent which can be used to tighten the bounds, but we believe that it does not allow us to understand the generalization phenomenon (see the paragraph before Section 4). To be convinced that the user-defined complexity measure is key for the bound's tightness, let $\mu(h,{\cal S})=\frac{\alpha}{m}\phi(R\_{\cal D}(h), R\_{\cal S}(h))-\alpha R\_{\cal S}(h)$ which directly integrate the generalization gap in the parametric function $\mu()$. Hence, if both hypotheses $h\sim \rho\_{\cal S}$ and $h'\sim\pi$ generalize well, the gaps $\phi(R\_{\cal D}(h'),R\_{\cal S}(h'))$ and $\phi(R\_{\cal D}(h),R\_{\cal S}(h))$ will be small (with high probability) providing a small generalization bound. Hence, the choice of the parametric function $\mu(h, {\cal S})$ is important and must reflect the true risk. We added this discussion after Corollary 3.1.
>
> *About the missing references.* Concerning the references, we extend the “Related works using the Gibbs Distribution” (i.e., the related works) section. In order to reply in the best way to your concerns, we add several discussions about the Gibbs distribution. First, we discuss the relationship between the information risk minimization principle and this distribution (introduced notably by [6]). Secondly, we extend the discussion on the information-theoretic generalization bounds [1,2] considering this distribution in [1,4,5]. Moreover, we add the reference [3] at the beginning of Section 2.2.
>
> _References:_
>
> [6] Tong Zhang. “Information-theoretic upper and lower bounds for statistical estimation.” IEEE Trans. Inf. Theory 52(4): 1307-1321 (2006)

---

### Official Review · Reviewer_m9hB · 2022-10-25

**Confidence:** 3
**Correctness:** 4
**Technical Novelty And Significance:** 4
**Empirical Novelty And Significance:** 4
**Recommendation:** 6

**Clarity, Quality, Novelty And Reproducibility:**

This is a highly technical yet relevant work. It operates at a very high level of abstraction, which makes understanding a little bit difficult. Nevertheless, I believe this to be an overall strength of the paper -- the entire point of the paper is that it applies to arbitrary complexity measures. I am not familiar with the literature in this area, but this work appears to be novel for me.

**Strength And Weaknesses:**

This paper offers an interesting general theoretical result that leverages its generality to increase its practical relevance. In particular, it allows a practitioner to understand their algorithms convergence for arbitrary complexity measures. This is particularly relevant because it is well known that traditional complexity measures often do not provide relevant bounds for practical forms of generalization.

I think this paper could have included a further discussion of complexity measures -- both including more examples as well as providing more context to their formalism. In particular, I believe that outlining common properties that one would expect the function $\mu$ to have could be very helpful for building better intuition.

**Summary Of The Paper:**

This paper provides a PAC-bayes bound that is able to utilize and arbitrary complexity measure. They define a complexity measure, $\mu$ as a function that maps a pair $(h, S)$ to a real number, where $h \in \mathbb{H}$ is a hypothesis, and $S$ is a labeled training sample. This notion generalizes common complexities such as the norm-weight, or the Rademacher complexity. They then show that the following posterior distribution, $\rho_S(h) \propto \exp \left[ -\alpha R_S(h) - \mu(h, S)\right]$ has a generalization gap that can be bounded with high probability. Their bound is highly general, due to the abstraction of $\mu$.

To empirically validate their results, they also give a way of approximately sampling from the posterior distribution above (which is quite non-trivial due to the nature of the distribution). They then apply this to several natural complexity  measures.

**Summary Of The Review:**

I recommend this paper for acceptance. I think that it could benefit from increased intuition and more examples, but both of these are easily addressable.

---

> ### Author Response · Authors · 2022-11-19
> **Answer to Reviewer m9hB**
>
> Thank you for your review and your feedback.
>
> *About desirable properties.* A key point is to integrate into the complexity measures some key information for assessing the generalization gap: this can be (norm) functions on the parameters, some (numeric) properties of the model, and/or some additional elements like the empirical error on another dataset. In Appendix E.2, we have notably proposed other examples of complexity measures, including some information on model parameters accompanied by an empirical risk on augmented data. The definition is very flexible: we can imagine including the risk of a validation set or of a source sample if we are in a domain adaptation scenario. So the possibilities are numerous. That being said, suggesting the right elements to provide is difficult since, at the moment, generalization capabilities in machine learning are not fully understood in general. Anyway, having some information on the model that can be optimized when trying to minimize the bounds is obviously key.
>
> *About a discussion of complexity measures.* In order to increase the intuition of the function $\mu()$, we discuss the function more in Section 3.1.

---

### Official Review · Reviewer_g9Hx · 2022-10-26

**Confidence:** 3
**Correctness:** 2
**Technical Novelty And Significance:** 3
**Empirical Novelty And Significance:** 3
**Recommendation:** 5

**Clarity, Quality, Novelty And Reproducibility:**

The paper is generally well written. However, there are substantial issues with Section D of the supplementary (see weaknesses).  Would also like to hear from the authors regarding point 2 in the “weaknesses”.

I also personally found it a little hard to end up with a good grasp of what the three terms in Theorem 2.1 correspond to, but it may be because I have less exposure to the PAC Bayesian setting.


**Strength And Weaknesses:**

Strengths:

Disintegrated PAC Bayesian bounds are quite interesting and the main result theorem 3.1 seems original and worthwhile
I like the fact that the appendix always puts a copy of the theorems to prove, which means the reader doesn’t have to keep going back and forth between the paper and the appendix.
The writing is good in general and there are not many grammatical mistakes.
The proof of Theorem 3.1 appears correct to me, and so does the proof of Corollary 3.1.
The stochastic MALA algorithm seems to be correctly implemented.

Weaknesses (including main points to be addressed in a rebuttal, numbered for the authors’ convenience):

1.	(TL;DR: state the exact page of Nagaragan and Kolter where Proposition D.2 appears, substantially rewrite and clarify Section D)
The paper claims that their bounds supersume the uniform convergence approach, but It is very hard to make sense of the “proofs”. The claim is also misleading in general since no bound analogous to uniform convergence bounds is derived for any concrete architecture.  There are errors in the “definitions” as well. Definition 3.1 is not very well written, but it is still technically possible to make sense of it. On the other hand, Section D makes very little sense.
     1.1 **The “proofs” of propositions D.2 and D.3 (which are almost exactly identical) merely seem to go back and forth through tautologies**. Although I admit I may have missed something due to my lack of familiarity with PAC Bayesian bounds, I am quite certain that the proofs are at best very unclearly executed: Proposition D.2 seems to be a reformulation of the concept of uniform generalization bounds taken from Nagarajan and Kolter 19. The authors say that NK19 failed to provide a proof, but it seems unlikely that the result in NK19 needs proof and the current treatment is messy. **Prop. D.2 states that \Phi_u needs to fulfil definition D.1, which includes equation (14).  This seems to be an assumption of the proposition. The proposition statement than says that equation (14) (from the assumed definition D.1) is equivalent to another trivial reformulation of it.**

1.2	The nearly identical Corollaries D.1 and D.2 are not better. First of all, the corollaries claim to use Theorem 3.1 but do not explicitly state where they do so, though I believe it is exactly after the line “for any hypothesis h… the definition of the parametric function… bound”.  The conclusion there does not seem sound either: Theorem 3.1 is implicitly used there in a form that doesn’t involve the probability over h’ being drawn from the prior. It is also not clear to me why the definition of mu is different depending on whether h=h’ or not (please explain).
2.	In Section 4.2 (experiments), you claim that the set H  “corresponds to the hypotheses h_w that can be obtained from this initialization” (the initialization being that of He et al. 2015). This initialization requires drawing from gaussian distributions, but corollary 3.1 only applies to a uniform distribution on H. Am I correct in assuming that you use the scaling/normalizing factor of the normal initializations from He et al. 2015 to define lower and upper thresholds that define a uniform distribution over the weights? If so, this needs to be explained thoroughly. It is also very unclear how this would maintain the benefits of He initialization since uniformly sampling from a hypercube is radically different from sampling from a multivariate gaussian in terms of dimensional dependence.
3.	 It seems like most of the experiments are conducted at a regime where there is so much data that the generalization gap is near zero, which makes the bounds far less informative.


**Summary Of The Paper:**


A general framework to compute PAC-Bayesian Generalization bounds is provided. The bounds are “disintegrated” in the sense that they bound the generalization gap by a function which involves both the trained hypothesis h and another hypothesis h’ drawn from the prior distribution. The bounds hold with high probability over the joint draw of the sample, h’ and h, where h is obtained via the Gibbs distribution from classic PAC Bayesian bounds. Sampling from this distribution is analogous to performing empirical risk minimization with added randomness/regularization which depends on the definition of a regularizing function mu.

The main theorems are Theorem 3.1, which applies to any priori, and Corollary 3.1, which applies to the case of a uniform prior over a set \mathcal{H}. Theorem 3.1 bounds the generalization gap with high probability by the sum of three terms: the last term is a function of delta and is similar to an expected generalization gap of a sample from the prior (modulo exponentials and logs): in more concrete situations such as Corollary 3.1, this term is O(constant/\sqrt{m}\delta^2) and therefore doesn’t actually depend on the prior pi. The control of this quantity in Cor 3.1 is obtained via a result of Maurer (cf. appendix F). The second term in Theorem 3.1 is the log of the ratio of the prior probabilities of h’ and h, which is a disintegrated analog of the KL divergence which would appear in classic PAC Bayesian bounds. Finally, the first term in Theorem 3.1 is the difference between the regularized risks of h (“trained”, i.e. drawn from Gibbs) and h’, drawn from the prior. This term can be interpreted as an empirically computable complexity measure, and is the main term left in corollary 3.1. The main ingredient in the proof of Theorem 3.1 is Theorem 2.1 which is taken from Rivasplata et al.
Experiments show that the bounds provide a good control of the true risk at some data-rich regimes where the generalization gap is nearly non existent. Sampling from the Gibbs distribution is achieved via a “stochastic” (i.e. batch-version of the)  Metropolis Adjusted Langevin Algorithm. Further experiments investigate the effects of the regularization parameter alpha.

There is an attempt to show that the proposed setting supersumes the uniform convergence approach (as well as algorithm dependent approaches) to generalization bounds, but the arguments are not convincing at all.



**Summary Of The Review:**

I enjoyed some parts of the paper, including the main theorem and corollary. The paper is generally well written, but some definitions are imprecise. The comparison to uniform convergences bounds unfairly places the current work above them without providing a concrete framework which could allow one to derive bounds as concrete as uniform convergence ones (without having to sample from the prior or estimate other quantities). Section D in the supplementary is sloppy and repetitive. Overall, only the short proof of Theorem 3.1  counts as an original theoretical contribution, which is not much for as prestigious a conference as ICLR.

I apologize if I missed something as I am not an expert on PAC Bayesian bounds, I am looking forward to the thorough rebuttal and willing to change my score if the other reviews and the rebuttal warrant it.



=================*More minor* comments=================

1. I would really appreciate a more specific reference (exact page and theorem) for Theorem 2.1 from Rivasplata et al. A proof is provided in the present paper and it seems correct, but it would be nice to see where the result is from (I couldn’t find the exact statement easily when browsing through the reference).

2. In the proof of Theorem 2.1, you might want to explain how you get to the last line in terms of the change of measure which absorbs the $ln(pi(h’)/\rho_S(h’))$ term. This is especially important since there are other minor typos in the proof (\mathcal{S} and \mathbb{S} are used interchangeably to refer to the same quantity and $\phi(h,S)$ should be $\phi(h’,S)$ in the exponential at the end of the first equation.

3. On page 13 “by simplifying the left-hand side …” I think you mean “right-hand”. Note that the last two equations immediately above that line are exactly identical. It seems there is a missing simplification (it is not hard to fill the gap though).
It would be nice to have a pointer to the proof of Prop. D.1 in section D.2, since the proof is in section D.4.
In the preliminaries (“setting”), in the definition of the risk, it seems that you are restricting yourself to classification, which needs to be made clearer earlier on. It seems the assumption is necessary to define kl the quantities in Corollary 3.1.

4. The notation in the main Theorem 3.1 includes two uses of h’ as a different dummy variable.

   5.Below corollary 3.1, “kl” (the KL divergence of the Bernouilli distributions induced by the numbers (\in (0,1)) provided as arguments ) is not defined. A pointer to page 27 would be nice.

6. For readability, it might be nice to repeat the fact that the quantity at line six of the algorithm can be computed despite the fact that \rho_U cannot, due to the simplification of the normalization constants in the numerator and denominator.

7. Do you know whether there are guarantees analogous to those of Chib and Greenberg for the stochastic variant of MALA you use? Is the stochastic version original to your work or does it appear elsewhere?


=======================Very minor typos etc.======================


1. Capital letter at “Theorem 3.1” on line two of the proof of Corollary 3.1
There are a few somewhat inelegant sentences such as:

 2. Section 2.2 just above definition 2.1: “obtained after obtaining”

  3.  Page 5: “Concerning the tightness of the bounds, it may appear loose.”

   4. Top of  Page 15: “ The generality of our framework can thus generalize”

---

> ### Author Response · Authors · 2022-11-19
> **Answer to Reviewer g9Hx**
>
> *About weakness 1.* In order to clarify section D, we perform an in-depth rewriting and simplification of the text and the proofs. To do so, we remove the set-theoretic viewpoints of the bounds detailed in Propositions D.2 and D.3 that were not necessary to prove the corollaries (that we merged in the new section). Indeed, the two propositions seem very confusing and might address your point 1.1. Concerning point 1.2, we mentioned Theorem 3.1 in the proof of the new corollary.
>
> *About weakness 2.* We think that there is a misunderstanding here: the initialization procedure follows a multivariate uniform distribution that is defined through He et al.'s method. We would like to point out that He et al. (2015)'s method can also be applied to multivariate uniform distributions, even though in their paper, a Gaussian distribution is considered. See, for instance, the file NiN_NN.py (lines 40 to 48) how it is implemented, which is based on PyTorch's “torch.nn.init.kaiming_uniform_” function. We choose to focus on uniform distribution for the prior in Corollary 3.1 for simplicity reasons since it can be a classic choice in the PAC-Bayesian theory; note, nevertheless, that our framework can also be adapted to other distributions such as Gaussian distributions.
>
> *About weakness 3.* You are right; in this regime, i.e., with enough data points, the generalization gap might be small. While our bound is not the tightest in practice, in theory, it has the possibility nevertheless to be tight when $\mu(h,{\cal S})=\frac{\alpha}{m}\phi(R\_{\cal D}(h), R\_{\cal S}(h))-\alpha R\_{\cal S}(h)$.
> Indeed, the bound for $\phi(R\_{\cal D}(h), R\_{\cal S}(h)) = m\text{kl}[R\_{\cal S}(h)\\|R\_{\cal D}(h)]$ becomes $[\tfrac{1}{\sqrt{m}}\left(\text{kl}[R\_{\cal S}(h’)\\|R\_{\cal D}(h')]{-}\text{kl}[R\_{\cal S}(h)\\|R\_{\cal D}(h)]\right){+}\frac{1}{m}\ln\frac{2\sqrt{m}}{\delta}]\_+$ which is tight (with high probability) since the hypotheses with small gaps are likely to be sampled. Note also that the prior distribution can depend on a portion of the data. Such priors are widely used in the PAC-Bayesian theory (see, e.g., Pérez-Ortiz et al. 2021, in the context of neural networks) to tighten the bounds. However, using this technique is not helpful in understanding the generalization phenomenon: it only shows that the prior distribution is good but does not give any insight into why this prior allows sampling hypotheses that generalize well.
>
> *About the minor comments.*
>
> 1-2-3. We corrected the typos in the proofs; thank you for pointing them out.
>
> 4. We change the notation to clarify the proofs and the paper.
>
> 5. We added the definition in Section 3.3, i.e., in the main text of the paper.
>
> 6. Your suggestion has been added to the main text after the presentation of stochastic MALA.
>
> 7. To the best of your knowledge, we are unaware of works using a stochastic variant of MALA that we are using. Hence, we do not think that there are guarantees for this variant, and it can be addressed in future works.
>
> *About the very minor typos.* We modified the sentences that you suggested.
>
> Finally, we would like to thank the reviewer for his careful reading and relevant feedback. About the fact that only the (short) proof of Theorem 3.1 counts as an original contribution, we hope that the modifications made in Appendix are sufficient enough to strengthen the theoretical part. Due to the space limitation, we had to make some choices, and we thought it was a better idea to give more space to the framework in the main paper. We also would like to highlight again that we are not aware of existing general frameworks that are able to develop generalization bounds with arbitrary complexity measures, which we hope can also be seen as a valuable contribution.

---

### Author Response · Authors · 2022-11-19
**General Answer**

We would like to thank the reviewers for their time and constructive feedback. We believe that these reviews helped a lot to improve the paper. Please note that we revised the paper according to it; the main edits are highlighted in red.

We also would like to stress that the goal of our paper was not to provide the tightest bounds but to provide, to the best of our knowledge, the first generalization bounds able to incorporate user-defined complexity measures.

---

### Decision · Program_Chairs · 2023-01-20

**Decision:**

Reject

**Justification For Why Not Higher Score:**

Please see the above.


**Justification For Why Not Lower Score:**

N/A


**Metareview: Summary, Strengths And Weaknesses:**

The authors present a new, very general, bound on generalization, based on “disintegrated PAC bounds” presented in earlier work, and some empirical evaluation of the bound, which uses a new way to sample from a posterior.

The reviewers weren’t convinced that some key claims of the paper were rigorously supported.  The strength of the bounds was also not established convincingly.  The fundamental novelty of the analysis appeared limited.